# Determining Actual Evapotranspiration Based on Machine Learning and Sinusoidal Approaches Applied to Thermal High-Resolution Remote Sensing Imagery in a Semi-Arid Ecosystem

Luis A. Reyes Rojas [1], Italo Moletto-Lobos [2], Fabio Corradini [3], Cristian Mattar [2,*], Rodrigo Fuster [1] and Cristián Escobar-Avaria [1]

1    Laboratory of Territorial Analysis (LAT), University of Chile, Santiago 8820808, Chile; lreyesrojas@uchile.cl (L.A.R.R.); rfuster@uchile.cl (R.F.); crescobar@uchile.cl (C.E.-A.)
2    Laboratory for the Analysis of the Biosphere (LAB), Santiago 8820808, Chile; italo.moletto@um.uchile.cl
3    INIA La Platina, Instituto de Investigaciones Agropecuarias, Santiago 8831314, Chile; fabio.corradini@inia.cl
*    Correspondence: cmattar@uchile.cl

**Abstract:** Evapotranspiration (ET) is key to assess crop water balance and optimize water-use efficiency. To attain sustainability in cropping systems, especially in semi-arid ecosystems, it is necessary to improve methodologies of ET estimation. A method to predict ET is by using land surface temperature (LST) from remote sensing data and applying the Operational Simplified Surface Energy Balance Model (SSEBop). However, to date, LST information from Landsat-8 Thermal Infrared Sensor (TIRS) has a coarser resolution (100 m) and longer revisit time than Sentinel-2, which does not have a thermal infrared sensor, which compromises its use in ET models as SSEBop. Therefore, in the present study we set out to use Sentinel-2 data at a higher spatial-temporal resolution (10 m) to predict ET. Three models were trained using TIRS' images as training data (100 m) and later used to predict LST at 10 m in the western section of the Copiapó Valley (Chile). The models were built on cubist (Cub) and random forest (RF) algorithms, and a sinusoidal model (Sin). The predicted LSTs were compared with three meteorological stations located in olives, vineyards, and pomegranate orchards. RMSE values for the prediction of LST at 10 m were 7.09 K, 3.91 K, and 3.4 K in Cub, RF, and Sin, respectively. ET estimation from LST in spatial-temporal relation showed that RF was the best overall performance ($R^2$ = 0.710) when contrasted with Landsat, followed by the Sin model ($R^2$ = 0.707). Nonetheless, the Sin model had the lowest RMSE (0.45 mm d$^{-1}$) and showed the best performance at predicting orchards' ET. In our discussion, we argue that a simplistic sinusoidal model built on NDVI presents advantages over RF and Cub, which are constrained to the spatial relation of predictors at different study areas. Our study shows how it is possible to downscale Landsat-8 TIRS' images from 100 m to 10 m to predict ET.

**Keywords:** evapotranspiration; surface temperature; semi-arid ecosystems; remote sensing; Landsat-8; Sentinel-2; NDVI

## 1. Introduction

Evapotranspiration has a key role as a component of the hydrological cycle in terrestrial ecosystems [1]. In the past decade, ET has become an element to consider in future climate change effects on the water cycle [2]. Besides, monitoring ET has relevance for assessing the hydrological cycle at different levels, such as irrigation, water resource quantification and use, weather forecast, and drought indexes [3]. Land surface temperature (LST) is an important variable in the energy balance equation of the Earth's surface and in the estimation of ET [4]. Satellite sensors do not directly measure ET; therefore, algorithms or models are developed for ET estimation [5,6].

The actual evapotranspiration (ETa) is generally predicted as a fraction of maximum evapotranspiration, which through an energy balance approach is calculated from remotely sensed LST [7]. Furthermore, some methodologies integrate this LST approach in their ETa estimation, such as the Operational Simplified Surface Energy Balance Model (SSEBop) that relies on the LST, and the reference evapotranspiration (ETo) for ETa modeling [7–9].

However, satellites have different sets of optical and thermal sensors, spatial resolutions, and frequency of data acquisition. Usually, higher temporal resolution satellites have lower spatial resolutions; therefore, combining and relating different satellite sensors measurements is necessary in order to obtain higher frequencies and resolutions in areas with contrasting land surfaces. The development of disaggregation of remotely sensed LST (DLST) [10] allows for the capture of greater spatial differences in LST, which become valuable in semiarid ecosystems with bare soil and vegetation variability at short distances [11]. The DLST methodologies can be used in surface energy balance models for ETa modeling at higher spatial resolutions [12]. Currently, machine learning algorithms, such as cubist (Cub) [13,14] and random forests (RF) [15], have been evaluated in downscaling LST, but not in semi-arid ecosystems. Linear models [16,17], algorithms of RF [18–21] and cubist [19,20], have been used successfully for DLST, arguing that the use of machine learning approaches in capturing non-linear outliers is less sensitive than using linear functions [22].

The ET applications of DLST has been used in monitoring crop water requirements during the growing season [23]. However, ET quantification using remote sensing monitoring might get affected by the abovementioned differences between highly contrasting areas with cultivated, low vegetation density, such as a desert [11]. The semi-arid ecosystem of the Atacama Desert is characterized by a lack of precipitation, low humidity, and low cloud coverage [24–26]. Furthermore, in the Copiapó valley, several water-demanding activities coexist, increasing the stress in water use and decreasing water availability [27]. These practices contribute to increased pressure over water use in the valley, and a clear analysis of the water balance should be considered when evaluating the sustainable use of water, food security, and decision making [28]. Therefore, looking for higher frequency estimation of ET by remote sensing at a higher resolution provided by Sentinel-2 images might be a proficient alternative to water assessment in an area where water conflicts may arise. The aims of this study were (i) to downscale LST from optical sensor and indices derived from Sentinel-2 at 10 m using observations from the Thermal Infrared Sensor (TIRS) of Landsat-8 using cubist, RF, and a sinusoidal model; and (ii) to estimate ETa using the DLST approach applied in the SSEBop model in an arid or semi-arid climate of the Copiapó valley.

## 2. Materials and Methods

### 2.1. Study Area

The area of study is in the Copiapó valley in the Atacama Region, Chile. The Copiapó river watershed is about 18,538 km$^2$, stretching from the Andes to the Pacific Ocean coast. The area of study is located in the nearest zone to the pacific coast (Figure 1). In this section of the valley, the water source is mainly from aquifers, extracted through wells and applied with high-frequency irrigation systems [29] (Figure 1). The agriculture in this sector is dominated by olive trees, table grape orchards, pomegranates, tomatoes, and natural vegetation. The climate is semiarid to arid, with hot and dry summer seasons, and 28 mm of mean annual precipitation [27,30].

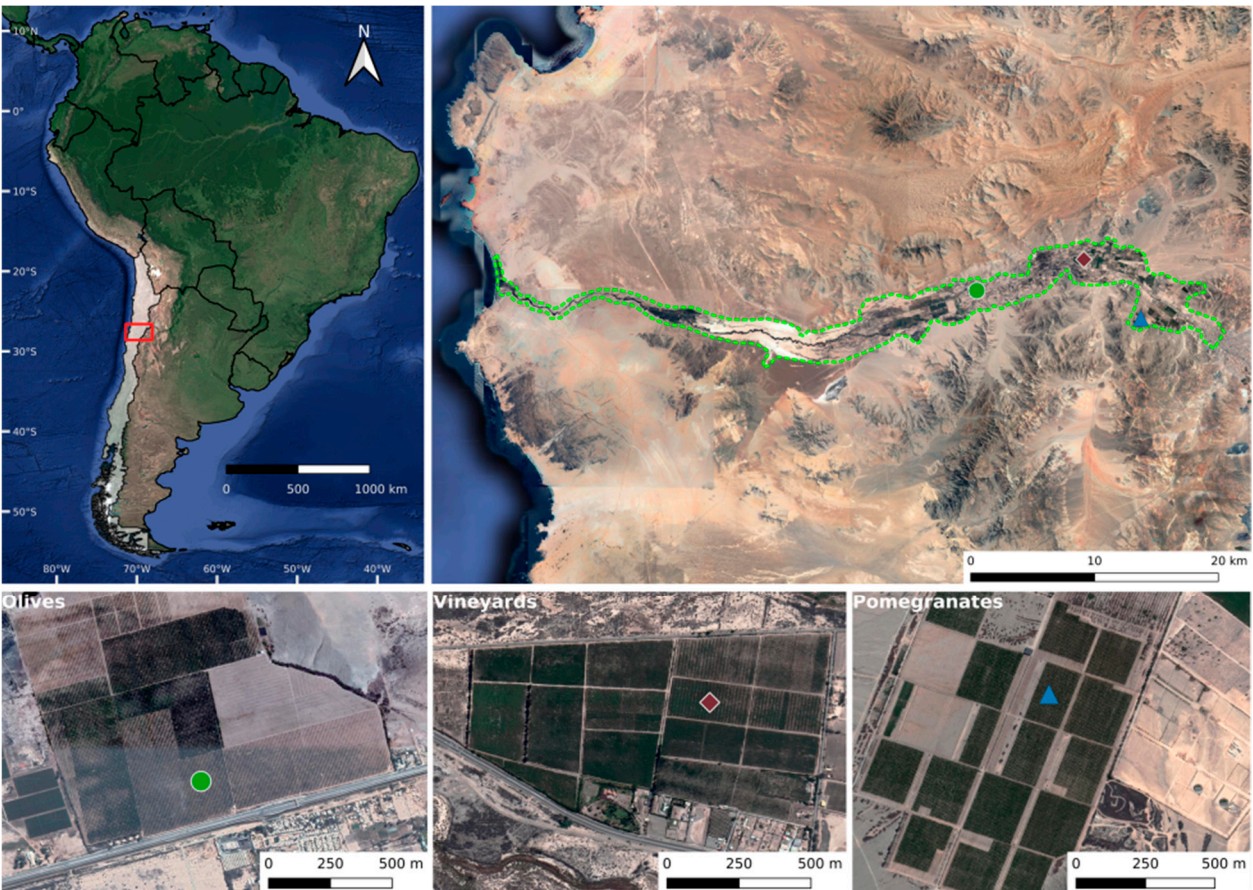

**Figure 1.** Study area delimited by a green light dashed line (top). Location of three meteorological stations distributed in olives (green circle), vineyards (red diamond), and pomegranates (blue triangle).

### 2.2. Local Data

Meteorological stations data were obtained from LAB-network [31], located across the valley over three crops, as was described in Mattar et al. [31] and Olivera-Guerra et al. [27]. Crop coefficients (kc) used are in concordance with those used by Olivera-Guerra et al. [27] for olives and vineyards (Table 1), and the pomegranates kc values were adapted from Franck [32] and Otárola Aliaga [33], which estimated kc values to arid and semiarid conditions in Chile. The stations located in olives and vineyard orchards have a continuity of in situ data from January 2016, while the pomegranates station started to measure in the winter of 2019, all orchards are under drip irrigation.

**Table 1.** Olives, vineyards, and pomegranates kc values.

| Crop | Jan | Feb | Mar | Apr | May | Jun | Jul | Aug | Sep | Oct | Nov | Dec | References |
|------|-----|-----|-----|-----|-----|-----|-----|-----|-----|-----|-----|-----|------------|
| Olives | 0.65 | 0.65 | 0.65 | 0.65 | 0.6 | 0.5 | 0.5 | 0.5 | 0.6 | 0.6 | 0.65 | 0.65 | [27] |
| Vineyards | 0.7 | 0.65 | 0.6 | 0.5 | 0.4 | 0.4 | 0.4 | 0.4 | 0.4 | 0.6 | 0.65 | 0.7 | [27] |
| Pomegranates | 0.6 | 0.68 | 0.8 | 0.45 | 0.4 | 0.115 | 0.115 | 0.3 | 0.3 | 0.4 | 0.4 | 0.45 | [32,33] |

### 2.3. Remote Sensing Data

The product used for remote sensing was the Level 1-C Top of Atmosphere Reflectance (TOA) and Surface Reflectance from Landsat 8 Level 2-A product. Thermal data were obtained from band 10 digital numbers (ND) of Landsat 8 Thermal Infrared Sensor (TIRS) for the 2016–2020 period. Landsat-8 and Sentinel-2 images were cloud masked using the Fmask 4.0 algorithm [34], and spatially matched according to the study area and study

period. There were 27 dates that match between Landsat-8, and Sentinel-2 used for model calibration and validation. For each match date of Sentinel-2 and Landsat-8 TIRS images, 18 pairs of images were used in calibration of the Cub, RF, and sinusoidal (Sin) models, and 9 pairs of images for validation of the LST were estimated.

*2.4. Methodology*

2.4.1. Surface Reflectance Retrieval

The Sentinel 2 Level 2-A Surface Reflectance product was retrieved using a sen2cor processor [35], which corrects the image using Water Vapour (WV) and Aerosol Optical Thickness (AOT). This method performs atmospheric correction using Look-Up tables from libRadtran [36] using as baseline the mid-latitude summer (MS) for the aerosol and water vapour concentration for the study area. Water Vapour is obtained with Atmosphere Pre-Corrected Differential Absorption algorithms [37] using the B8A and B9 bands for reference channels in the atmospheric window and absorption region, respectively. The Aerosol Optical Thickness is derived from 550 nm using the Dense Dark Vegetation (DDV) algorithm [38], which correlates B12 versus VIS (B2, B3, B4). In order to determine the impact atmospheric inputs of the imagery, we assessed the mean value, standard deviation and coefficient of variation (cv) per pixel of WV and AOT for the study period. We evaluated the impact of topographic illumination relating the hillshade of every image versus the bands and NDVI determining the coefficient of determination of the imagery.

2.4.2. Land Surface Temperature Retrieval

The Landsat-8 LST was determined by a single-channel algorithm using the band 10 [39], which is defined as

$$\text{LST} = \gamma \left[ \frac{1}{\varepsilon} (\varphi_1 \cdot \text{L}_{\text{sen}} + \varphi_2) + \varphi_3 \right] + \delta \tag{1}$$

where $\gamma$, $\delta$ are two parameters that depend on the at-sensor brightness temperature, $\varepsilon$ is surface emissivity, and $\varphi_1$, $\varphi_2$, $\varphi_3$ are the atmospheric functions versus atmospheric water vapor content [40,41]. These input data of the atmospheric functions were obtained by polynomial equations proposed by Cristóbal et al. [40] using NCEP/NCAR Reanalysis data, which models the ascending, descending, and ascending atmospheric radiance and transmittance (Lup, Ldown, and T, respectively). The $\varepsilon$ input was obtained from the ASTER Global Emissivity Dataset (ASTER GED) [42]. Thermal radiance (Lsen) was obtained from radiometric calibration of band 10. This LST obtained from Landsat-8 will be used as the observed data, and Sentinel-2 bands and spectral indices were considered as predictors for the LST modeling.

2.4.3. Predictors

The dataset used as predictors were 13 bands from Sentinel-2 plus 22 remote sensing indices (Table 2), which were used as calibration data for the Cub and RF methods. The pixel resolution for Landsat-8 LST and NDVI was 100 m as the observation data, and 10 m for the Sentinel-2 bands and indexes. Sentinel-2 data were resampled to 100 m at each date for modeling as predictors at 100 m and 10 m. The range of images acquisition was from February 2016 to January 2019 in the calibration data, and from April 2019 to April 2020 in the validation data in the matching dates, and Sentinel-2 data between February 2016 and June 2020.

**Table 2.** Set of sentinel bands and indexes tested for LST modeling.

| Name Variables | Sentinel-2 Variables | Expression | References |
|---|---|---|---|
| B1 | Band 1 | | |
| B2 | Band 2 | | |
| B3 | Band 3 | | |
| B4 | Band 4 | | |
| B5 | Band 5 | | |
| B6 | Band 6 | | |
| B7 | Band 7 | | |
| B8 | Band 8 | | |
| B8A | Band 8A | | |
| B9 | Band 9 | | |
| B11 | Band 11 | | |
| B12 | Band 12 | | |
| NDVI | Normalized difference vegetation index | $\frac{NIR-Red}{NIR+Red}$ | [43] |
| SAVI | Soil adjusted vegetation index | $\frac{NIR-Red}{NIR+Red+L} \times (1+L)$ | [44] |
| EVI | Enhanced vegetation index | $G \times \frac{NIR-Red}{NIR+C1\times Red-C2\times+L}$ | [45] |
| GNDVI | Green normalized difference vegetation index | $\frac{NIR-Green}{NIR+Green}$ | [46] |
| NDWI | Normalized difference water index | $\frac{Green-NIR}{Green+NIR}$ | [47] |
| MSAVI2 | Modified soil vegetation index 2 | $\frac{2\cdot NIR+1-\sqrt{(2\cdot NIR+1)^2-8\cdot(NIR-Red)}}{2}$ | [48] |
| ALBEDO | Albedo | $\alpha = \sum_{B\,i}|\rho_{B\,i}\cdot\omega_{B\,i}|$ | [49] |
| SELI | Sentinel-2 LAI$_{green}$ index | $\frac{B8a-B5}{B8a+B5}$ | [50] |
| TCARI | Transformed chlorophyll absorption ratio index | $3\cdot((B5-B4)-0.2\cdot(B5-B3)(B5/B4))$ | [51] |
| OSAVI | Optimized soil adjusted vegetation index | $\frac{(1+0.16)(NIR-Red)}{NIR+Red+0.16}$ | [52] |
| TCARI/OSAVI | | $\frac{TCARI}{OSAVI}$ | [51,52] |
| GRVI | Green-Red vegetation index | $\frac{Green-Red}{Green+Red}$ | [53] |
| WDRVI | Wide dynamic range vegetation index | $\frac{0.1\cdot NIR-Red}{0.1\cdot NIR+Red}$ | [54] |
| BWDRVI | Blue-wide dynamic range vegetation index | $\frac{0.1\cdot NIR-Blue}{0.1\cdot NIR+Blue}$ | [55] |
| TVI | Transformed vegetation index | $\sqrt{NDVI+0.5}$ | [43] |
| ARVI | Atmospherically resistant vegetation index | $\frac{NIR-Red-y(Red-Blue)}{NIR+Red-y(Red-Blue)}$ | [56] |
| SIPI | Structure insensitive pigment index | $\frac{B8-B1}{B8-B4}$ | [57] |
| BSI | Bare soil index | $\frac{(SWIR+Red)-(NIR+Blue)}{(SWIR+Red)+(NIR+Blue)}$ | [58] |
| MSI | Sentinel-2 Moisture stress index | $\frac{B11}{B8}$ | [59] |
| GCI | Green chlorophyll index | $\frac{B9}{B3}-1$ | [60] |
| NDMI | Normalized difference moisture index | $\frac{NIR-SWIR}{NIR+SWIR}$ | [61] |
| CLRE | Red-edge-band Chlorophyll Index | $\frac{B9}{B5}-1$ | [60] |

G, C1, C2: Coefficients; NIR: Near infrared; SWIR: short wave infrared; $\rho_{Bi}$: surface reflectance at band Bi; $\omega_{Bi}$: weighting coefficient at band Bi.

### 2.4.4. Spatial Relationship between Landsat-8 LST and Sentinel-2

The spatial relationship between Landsat-8 LST and predictors from Sentinel-2 were by two machine learning algorithms, cubist [13,14] and random forests [15,62], and one through a sinusoidal relationship between LST and NDVI [27], based on Bechtel et al. [16] and Bechtel [63,64]. For building each model, the LST at 100 m is the target variable, excepting NDVI at 100 m, which is also needed for the sinusoidal model. The Sentinel-2

images at 100 m are used to predict LST at 100 m according to each model; later, each Sentinel-2 at 100 m calibrated model was used to predict LST using 10 m Sentinel-2 predictors.

A cubist model is a tree of rules limited by conditions based on values or ranges of predictors. Each rule has a linear model that predicts the target value of a pixel in that condition. In this approach, the 27 calibration dates were spatially matched using the whole set of predictors using the *Cubist* package [65] in *R open source software* [66].

A random forest model consists of many decision trees that use several random subsampling creating a learning model based on classification or regression trees. The same 27 calibration dates of Sentinel-2 images were applied for prediction targeting LST from Landsat-8 with the *randomForest* package [67,68]. Furthermore, the *variable selection for random forest* algorithm (VSURF) was applied to allow parsimony and evaluate which predictors performed better predicting LST with random forests. The VSURF algorithm is implemented in R software by Genuer et al. [69,70]. Due to its high computational demand, the algorithm was run in Wageningen University's High Performance Computing Cluster (HPC), Anunna. The process involved 25 random samples of 11,000 points from the whole calibration dataset per each VSURF run. Then, 50 RF with 2000 trees each were run at each sample, and then the results were ranked by variable importance averaging of 50 RF runs [69]. Later, VSURF has three outcomes of the selected variables: thresholding, interpretation, and the prediction step. We chose interpretation, because it involves more variables than a prediction step, and it reduces overfitting from the thresholding step. Finally, the sum of the variables selected from those 25 VSURF runs were used as the set of variables in the RF prediction.

The sinusoidal modeling is based on a general linear relationship between LST and NDVI [71]:

$$LST_{10m} = a + b \cdot NDVI_{10m} \tag{2}$$

Furthermore, this relation is seasonal during the year, allowing estimation of annual cycle parameters according to Bechtel et al. [16] and Bechtel [63,64]. The relationship between LST and NDVI was validated for DLST in the study by Olivera-Guerra et al. [27].

$$LST_{L8\ 100m} = c_i + d_i \cdot NDVI_{S2\ 100m} \tag{3}$$

where $c$ is intercept and $d$ is the slope from the fitted linear values of Landsat-8 LST and Sentinel-2 NDVI at each $i$-calibration date. Therefore, the linear coefficients $c$ and $d$ can be modeled as annual cycle parameters depending on the day of the year relative to the spring equinox.

$$c = e + f \cdot \sin\left(\frac{DOY\ as\ equinox \cdot 2\pi}{365}\right) \tag{4}$$

$$d = g + h \cdot \sin\left(\frac{DOY\ as\ equinox \cdot 2\pi}{365}\right) \tag{5}$$

where $c$ and $d$ are linear coefficients of the LST–NDVI relationship to each image date. The $e, f, g$, and $h$ are fitted coefficients of the relationship between the day of the year with $c$ and $d$. The spring equinox is 21 September in the southern hemisphere, with a value of 0 to that day of *DOY as equinox*; therefore, $-182.5 \leq DOY\ as\ equinox \leq 182.5$. These $c$ and $d$ coefficients from the calibration dates were used for estimating $a$ and $b$ from Equation (2) using the fitted values of $e, f, g$, and $h$ for a 10 m resolution.

### 2.4.5. Estimation of Actual Evapotranspiration

The ET estimation method was the Operational Simplified Surface Energy Balance (SSEBop) [9]. The SSEBop model has been widely applied at different hydroclimatic regions [7], which is based on the estimation of the ET fraction (ETf), which is the ratio of latent heat flux to the Net Radiation. The ETf is retrieved using surface temperature ($T_s$), cold/wet ($T_c$), and hot/dry ($T_H$) idealized surface temperature conditions from

Bastiaanssen et al. [72]. Thus, ETf is calculated using the following equations from Senay et al. [9]:

$$ETf = \frac{T_H - T_s}{T_H - T_c} = \frac{T_H - T_s}{dT} \tag{6}$$

$$dT = \frac{R_n \cdot r_{ah}}{\rho_a \cdot C_P} \tag{7}$$

where dT is the difference in surface temperature between the idealized conditions and is calculated under clear-sky conditions and is unique for location and day of the year (DOY), but with the assumption of not changing from year to year [8]. $r_{ah}$ is the aerodynamic resistance to heat in an idealized bare and dry surface (s m$^{-1}$), which was used the value of 110 sm$^{-1}$ from Senay et al. [9]; $\rho_a$ is the air density estimated by a function of air pressure and the virtual temperature ($T_{kv}$) [73]. $C_p$ is the specific heat at a constant air pressure (1.013 kJ kg$^{-1}$ K$^{-1}$). $T_H$ is calculated by $T_H = T_c + dT$, and Tc is calculated by the relation of the maximum air temperature and a correction factor of 0.093 ($T_c = 0.093\ T_{max}$) [9,27]. The $R_n$ is clear-sky net radiation (W m$^{-2}$), and $R_n$ was obtained calculating the ratio between daily net radio and instantaneous radiation ($C_{di}$) in the function of DOY from Sobrino et al. [74], adapted by Moletto-Lobos [75] to the southern hemisphere:

$$C_{di} = \begin{cases} DOY \geq 183 \Rightarrow -7\cdot10^{-6}(DOY - 183)^2 + 0.0027(DOY - 183) + 0.124 \\ DOY < 183 \Rightarrow -7\cdot10^{-6}(DOY + 183)^2 + 0.0027(DOY + 183) + 0.124 \end{cases} \tag{8}$$

Then ETf is calculated using the $T_s$ derived from the downscaled LST of Cub, RF, and Sin method. ETa is calculated using ETf, the maximum crop coefficient (kc) during the phenological season by an aerodynamically rougher crop of 0.65 according to Olivera-Guerra et al. [27,76], and the calculated evapotranspiration of reference (ETo) using the standardized Penman–Monteith equation [73,77]:

$$ETa = ETf \times kc \times ETo \tag{9}$$

The ETo was calculated using the atmospheric inputs from the ERA5 product [78]. In previous studies using the SSEBop model by Senay et al. [8], they have defined those negative values of ETf are set to zero and maximum ETf are capped at 1.05. According to Senay et al. [7], ETf should vary between 0 and 1; therefore, in this study negative ETf were set to zero and capped to 1.

2.4.6. Validation In Situ

The validation of the downscaled LST and measured Landsat-8 LST were with the in situ LAB-net stations, and their thermal infrared sensor (Apogee SI-111®). This sensor is located at 5 m height with the inclination to measure the same fraction of vegetation cover of the terrain. The values of LST were extracted from the downscaled images to compare them with in situ stations. The in situ value were obtained by the mean of the surrounded pixels at the position of the corresponding olive, vineyard, and pomegranate stations. For the comparison of in situ ET, the crop evapotranspiration (ETc) was calculated using the ET calculated by the weather station and multiplying the monthly corresponding kc of each crop (ETc = ETo ·kc), following the coefficients shown in Table 1. The performance metrics were the root mean square error (RMSE), standard deviation (sigma), bias, and correlation coefficient (r) for the nine LST validation dates that match between Landsat-8 and Sentinel-2.

## 3. Results

### 3.1. Atmospheric Inputs and Topographic Variation

The atmospheric inputs of Sentinel 2 correction are shown in Supplementary Figure S1 for AOT. The mean value shows values related to clear sky places, such as the Atacama Desert, with a mean value of 0.17 and standard deviation of 0.01, without variability over

the time series. The WV (Supplementary Figure S2) had a mean value of 1.13 cm and coefficient of variation of 13.4%. The atmospheric inputs show an artifact due to the method of calculation on the center of study area; however, the low values of atmospheric water vapor do not change over values of an atmospheric profile for a semi-arid ecosystem. On the other hand, the coefficient of determination of hillshade of every image versus the input bands does not show any relation due to the flat terrain in the study area, with an average slope of 3%. The part that reaches the highest correlation is in the red band with an R2 of 0.2 in the place steeper area of 41%.

The atmospheric inputs of Landsat 8 are in Supplementary Table S1, where Ldown shows the most variation, with a maximum value of 5.70 W m$^{-2}$ sr$^{-1}$, minimum of 1.29 W m$^{-2}$ sr$^{-1}$, and coefficient of variation up to 28.41%. The Lup showed lower variability, with a cv of 16.06% and transmittance with cv of 9.42% and mean of 0.77 for the study period.

### 3.2. Cubist

The Cub output created a hundred rules for predicting LST, showing that almost all the predictors are used for prediction, but not all of them in defining the set of rules of the trees (Figure 2A). The most used variables in the algorithm to separate the conditions were B9 (all conditions) and then B11 and TCARI, participating in 63% and 53% of the rules, respectively.

### 3.3. Random Forest

The results of the VSURF algorithm (Figure 2B) show that in the 25 runs in the HPC computer, the variables selected for model interpretation after running 50 RF with 2000 trees per HPC run are B9, B11, B12, and TCARIOSAVI, which were selected in all VSURF runs, secondary CLRE in 23, and B1 in 20. The other variables were selected in a minor proportion but were included in the set of predictors used in the RF spatial prediction of LST at 100 m. Therefore, the RF model parsimony from VSURF that was used in the RF prediction of LST resulted in a total of 11 variables (Figure 2B,C). The variable importance as a percentage of increase in MSE, when one of these variables is out of the model, shows that the most important variables are B9, TCARIOSAVI, CLRE, and B1 (Figure 2C).

### 3.4. Sinusoidal

The sinusoidal model and observed fitted values according to the day of year relative to the spring equinox are shown in the Figure 2D,E, resulting in an intercept and slope modeling equation *c* and *d* of

$$c = 306.148 + 9.977 \cdot \sin\left(\frac{DOY \ as \ equinox \cdot 2\pi}{365}\right) \tag{10}$$

$$d = -14.118 - 5.047 \cdot \sin\left(\frac{DOY \ as \ equinox \cdot 2\pi}{365}\right) \tag{11}$$

The validation results (Figure 3) showed that for 100 m, the RMSE values were 5.77 K for Cub, 4.81 K for RF, 3.97 K for Sin models, and 6.4 K of Landsat-8 at 100 m. In terms of variation, the values of sigma followed the same previous trend with Cub with the higher values, followed by RF and then Sin. The RF and Cub models slightly overestimate and Sin underestimates LST, but also the correlation coefficient is the lowest for RF, followed by Cub and Sin with the best performance. In the 10 m prediction model, the trend in RMSE is similar: Cub performed with the highest RMSE 7.09 K, RF with 3.91 K, and 3.4 K for Sin. Besides, Cub shows a higher variation and overestimation of LST compared to the in situ values. RF at 10 m has a higher correlation coefficient and lower RMSE compared with 100 m. The Sin model shows an underestimation, the highest correlation coefficient, and lowest variation in the two-resolution LST estimation compared to in situ.

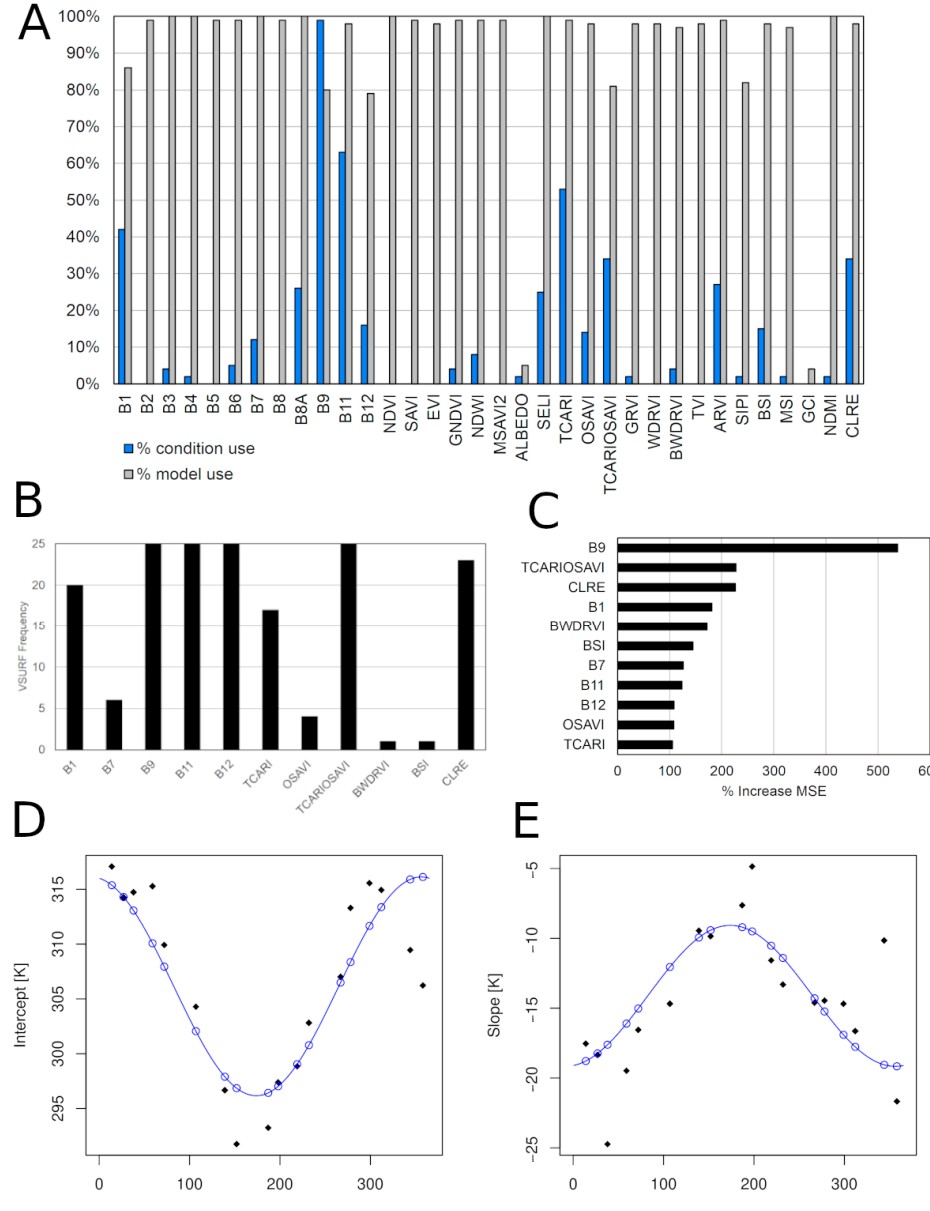

**Figure 2.** Cubist (**A**), variable selection for random forest algorithm (**B**), random forest (**C**), and sinusoidal (**D**,**E**) model calibration results.

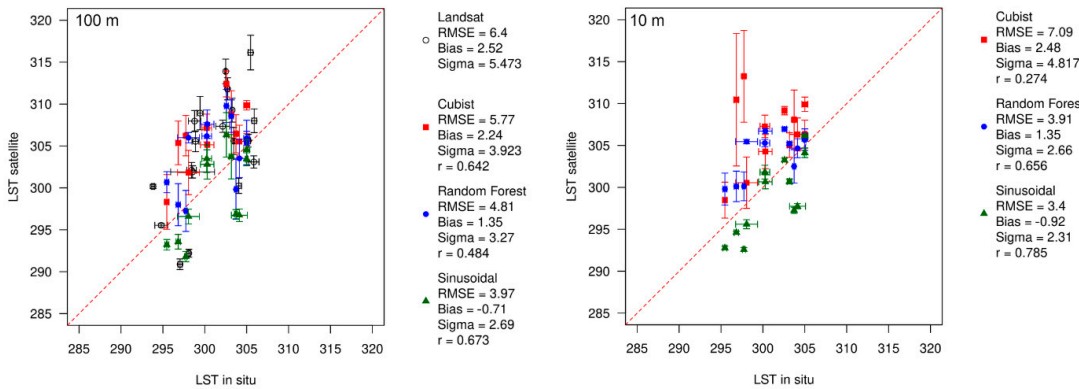

**Figure 3.** Validation results of the predicted LST with cubist, random forest, and sinusoidal models versus the LST measured by in situ stations.

The models are applied in a temporal series of LST at 100 m and 10 m (Figures 4 and 5, respectively) after validation dates in each crop, showing a general trend according to the bias validation results (Figure 3). For olives, the LST in all models is very similar to the LST trend during the seasons. However, Cub and RF overestimates the LST values of the station in winter and estimating in an opposite trend to the seasonal LST decrease in winter. The Sin model shows a tendency of seasonal variation of LST with a slight underestimation of LST, similar to bias validation results in Figure 3. In vineyards, the standard deviation of the surrounding pixels at the station of LST in Cub are higher than the other models and showed an increase in the winter LST estimated. The main difference in the 10 m resolution (Figure 5) is that Cub evidently has a higher standard deviation and is overestimating LST compared to the station in winter.

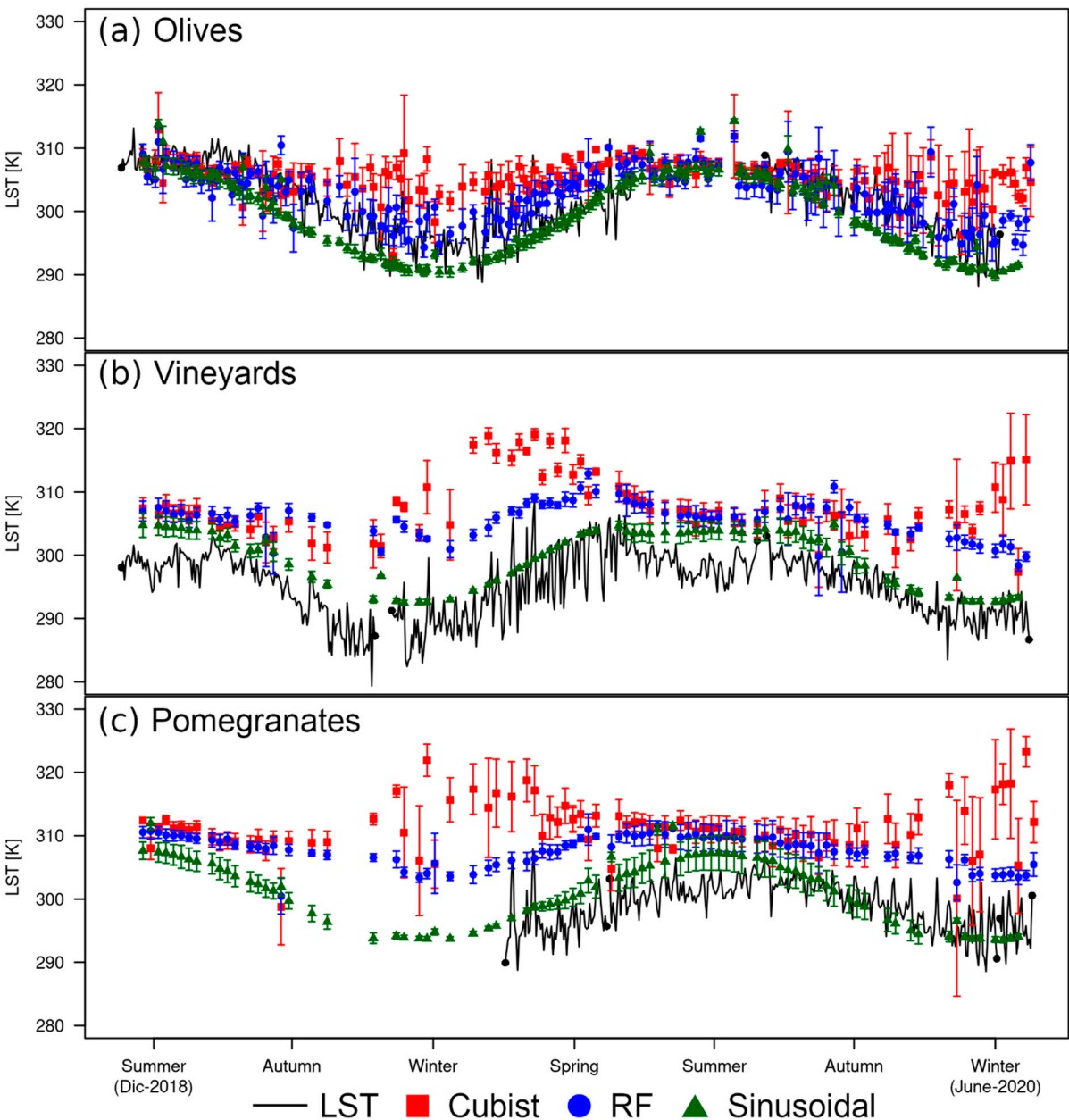

**Figure 4.** Temporal series of the predicted Sentinel-2 LST at 100 m of the cubist, random forest, and sinusoidal models compared to the LST measured by the in situ stations at (**a**) olives, (**b**) vineyards, and (**c**) pomegranates orchards. The error bars are showing the standard deviation of the 9 pixel cells surrounding the LST station of each LST model.

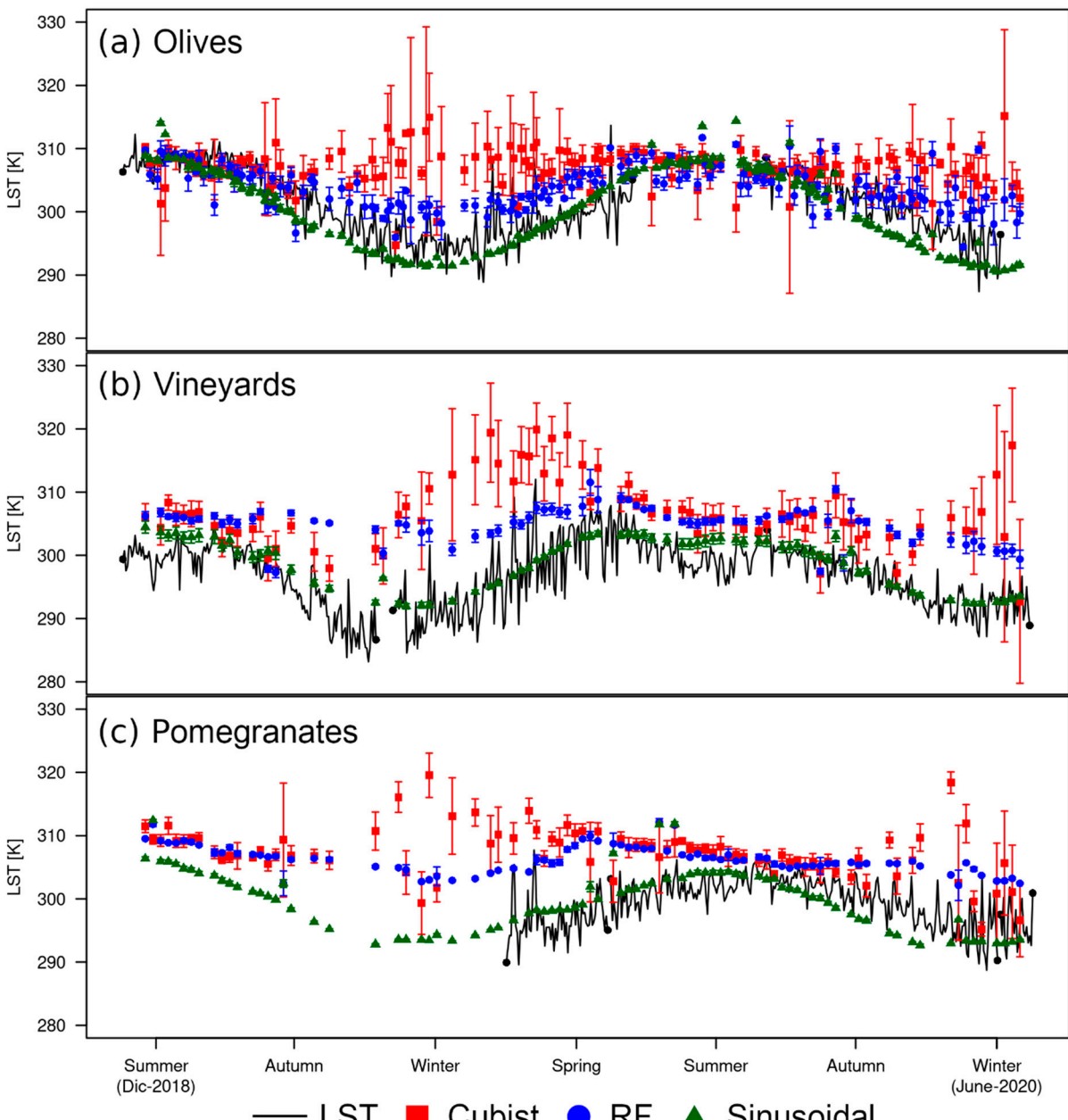

**Figure 5.** Temporal series of the predicted Sentinel-2 LST at 10 m of the cubist, random forest, and sinusoidal models versus the LST measured by the in situ stations at (**a**) olives, (**b**) vineyards, and (**c**) pomegranates orchards. The error bars are showing the standard deviation of the 9 pixel cells surrounding the LST station of each LST model.

The spatial prediction images (Figure 6) showed that in winter LST is higher for Cub and RF and lower for Sin at a 10 m resolution. Warmer pixels next to vegetation can be attributed to be bare soil captured by Landsat-8, which are colder in the Sin model. The Cub model shows a high variation in predicted LST pixels in a short range of spatial variation. Pixels varied between 280 to 290 K next to the warmest without a spatial relation to vegetation according to Landsat-8. In summer, all models showed spatially colder values than Landsat-8. The Sin model shows a clear distinction between vegetation areas, and RF does not show a clear spatial trend with vegetation, but the coldest pixels are related to areas with vegetation.

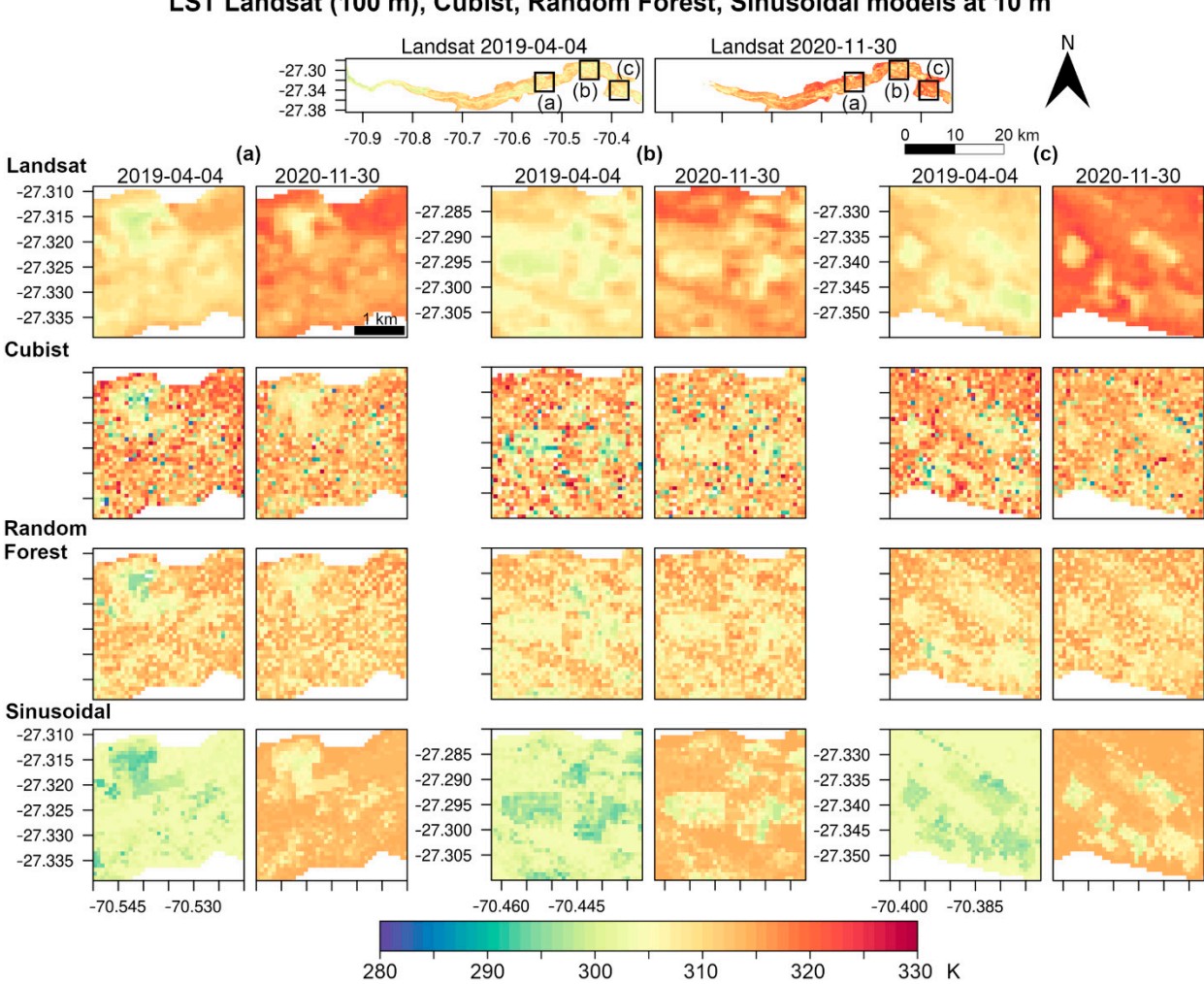

**Figure 6.** Land surface temperature by Landsat-8 at 100 m and the spatial predictions of LST from Sentinel-2 data using cubist, random forest, and sinusoidal models. The predictions are shown at two dates apr 04 2019 (winter) and nov 11 2020 (late spring) in three locations next to the olives (**a**), vineyards (**b**), and pomegranates (**c**) stations.

After being applied, the ETa spatial prediction of olives, vineyards, and pomegranates at 10 m were generated (Figure 7). The results showed winter (left) and a summer (right) images to each model of ETa per day, Cub and RF showed lower values of ETa compared to Sin in winter, and summer images are spatially similar in values with a clear distinction of crops by Sin. Although, with abrupt changes in closer pixels for Cub. In vineyards, a distinction is clear in winter for Sin, with higher values of ETa in the valley compared to Cub and RF. In the summer, the condition of short-range variation in ETa values in Cub continued similar to what was observed in the LST prediction, and RF and Sin also performed similarly. Pomegranate spatial values show a distinction of Sin in winter, showing higher values of ETa in the entire valley compared with Cub and RF. In the summer it performed similarly, but Cub continuously showed abrupt changes in a short range of pixels.

**Figure 7.** ETa based on Kmax at 10 m estimated from DLST with cubist, random forest, and sinusoidal models.

When the ETa temporal series is analyzed at 100 m (Figure 8) and 10 m (Figure 9), the values estimated for olives match almost entirely with ETc from the stations and kc values defined in Table 1. The Cub model shows an underestimation in winter, but all the other models followed the ETc seasonal variation in situ with an underestimation in summer. For pomegranates, the trend is the opposite, where generally all the models overestimate the ETc of station from winter to summer; however, in late 2019 summer to 2020 autumn, the estimated ETa followed the same trend as the station.

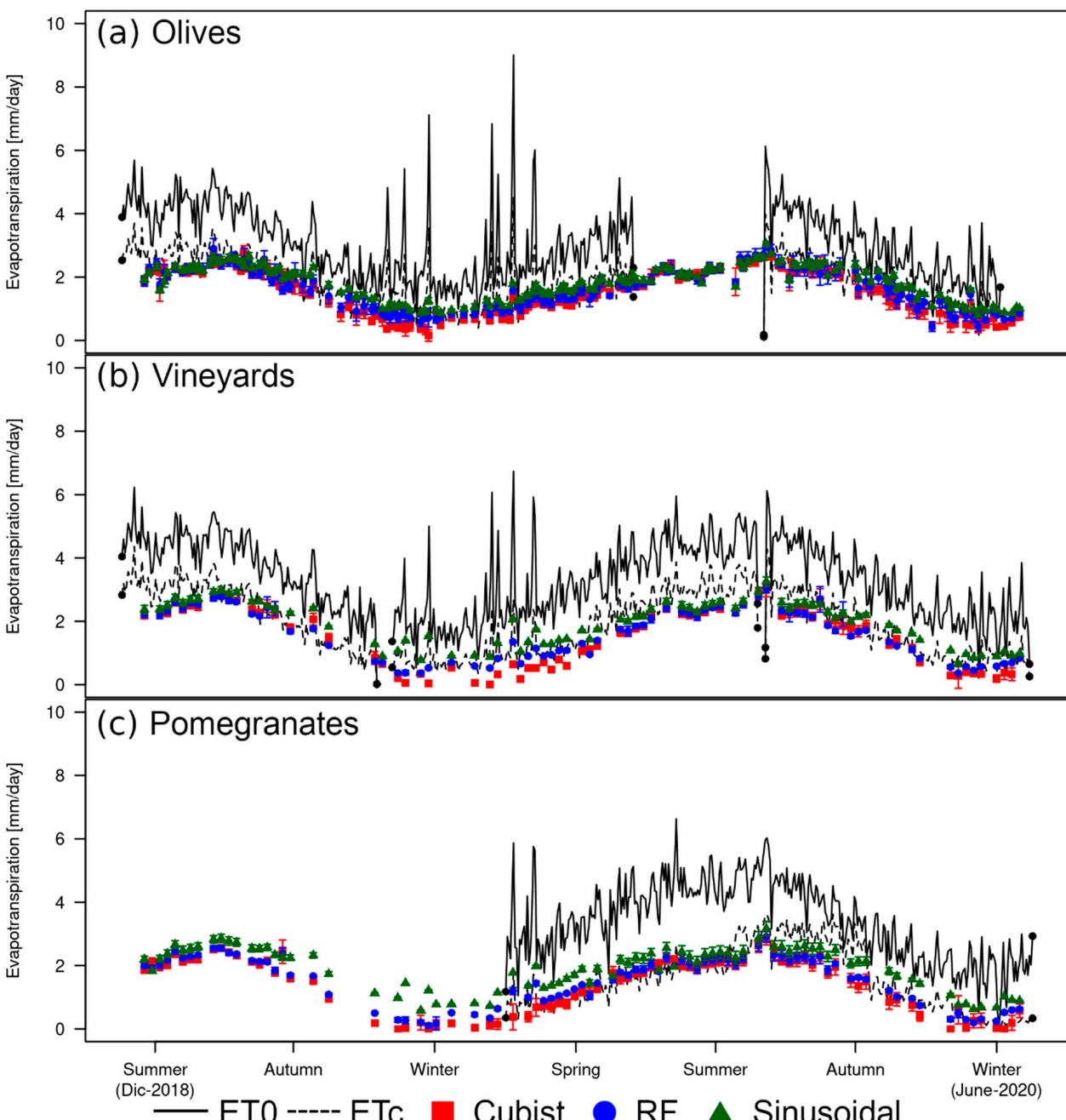

**Figure 8.** Temporal series of the predicted ETa at 100 m with cubist, random forest, and sinusoidal models versus the ETa measured by the in situ stations at (**a**) olives, (**b**) vineyards, and (**c**) pomegranates orchards. The error bars are showing the standard deviation of the 9 pixel cells surrounding the ETc station of each ETa model.

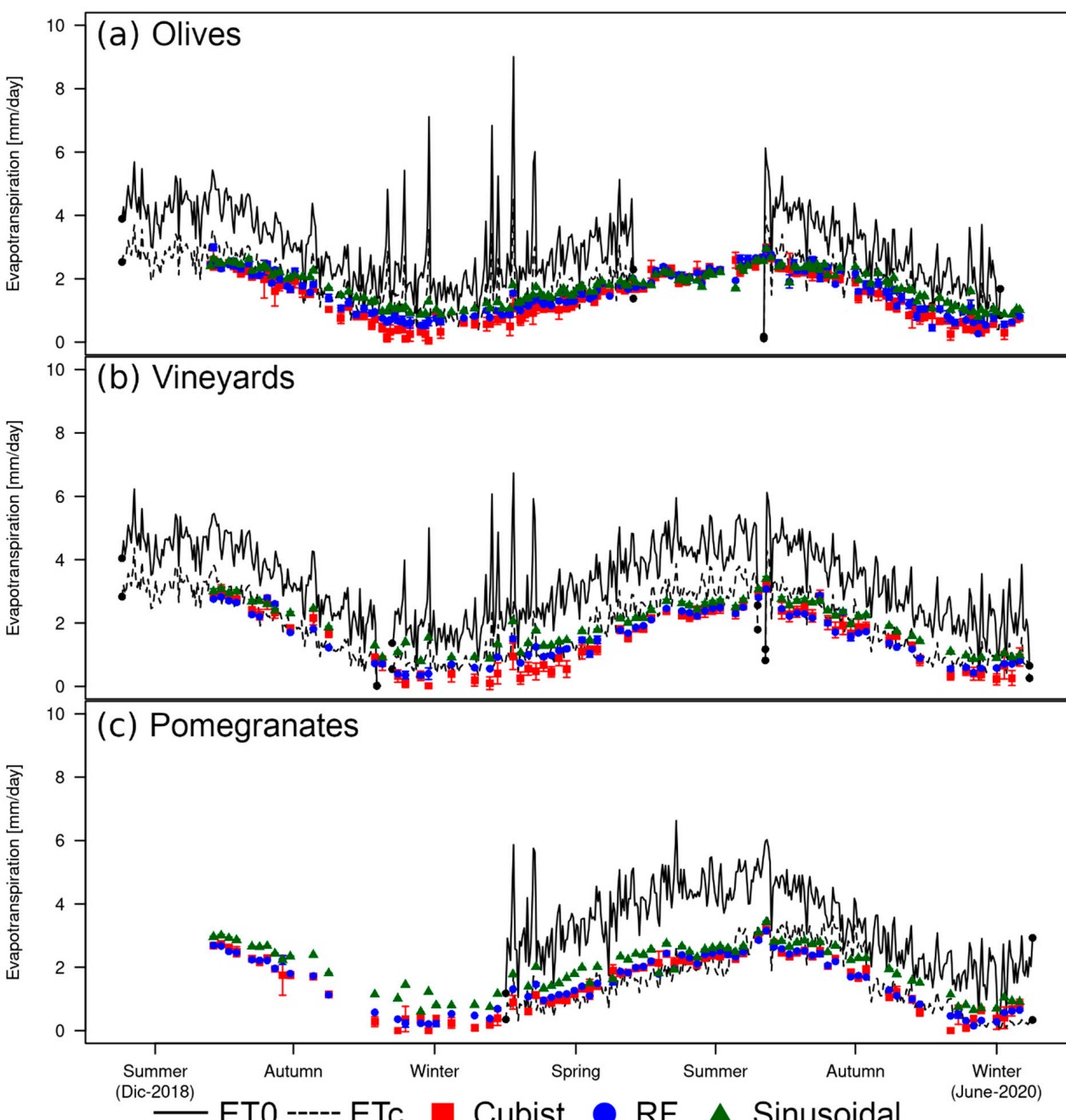

**Figure 9.** Temporal series of the predicted ETa at 10 m with the cubist, random forest, and sinusoidal models versus the ETa measured by the in situ stations at (**a**) olives, (**b**) vineyards, and (**c**) pomegranates orchards. The error bars are showing the standard deviation of the 9 pixel cells surrounding the ETc station of each ETa model.

## 4. Discussion

Testing and comparing new methods that quantify ET from irrigation crops is vital in areas with water scarcity, and detailed prediction maps allow a better decision-making process among water users [4]. The model applied for DLST in ETa based on Kmax are different spatially (Figure 7) and temporally (Figures 8 and 9), but the strength showed by the SSEBop ET model is consistent (Table 3), making these differences in LST predicted by each model (Figures 4–6) lower between Cub, RF, and Sin in ETa compared to ETc (Figure 10). The best model overall predicting ETa analyzed over ETc was RF with an $R^2 = 0.710$, Sin with 0.707, and an $R^2$ of 0.69 for Cub. However, the Sin model was with

the lowest RMSE of 0.45 mm d$^{-1}$, smallest bias, standard deviation, relative root mean square error (RRMSE), and mean absolute error. Besides, the Sin model is the best in olives and vineyards in all statistical indices, with the highest R$^2$ in all stations, but with the highest RMSE in pomegranates. The Cub model showed the lowest performance in all stations and overall analysis. This low ETa performance of Cub was noticed in the spatial ETa (Figure 7), with high variation of pixels over a short distance range. The performance of RF and Sin are consistent and similar in ETa at 10 m; nevertheless, there is a practical advantage of using the Sin model based in NDVI calibration compared with the RF model that is dependent on predictors to build a model by an empirical relation in one spatial area only. The meta-analysis obtained from the machine learning algorithms also gave approximations of a general approach estimating ET, which are evidenced by variables related with LST that showed high importance in the algorithms and may be important for improving future indices, equations, and models for calculating ET. However, the performance of the Sin model and its calibration process showed that it might be easier to apply with Sentinel-2 NDVI and without several calibration parameters that might be needed or differ for an RF model in a different region.

**Table 3.** Model performance statistics of ETa estimated using cubist, random forest, and sinusoidal models compared with the ETc over olives, vineyards, and pomegranates orchards.

| ETa | | Cubist | Random Forest | Sinusoidal |
|---|---|---|---|---|
| Olives | RMSE | 0.75 | 0.56 | 0.39 |
| | Bias | −0.35 | −0.25 | −0.12 |
| | Sigma | 0.56 | 0.42 | 0.29 |
| | R$^2$ | 0.673 | 0.750 | 0.798 |
| | RRMSE | 29.21 | 16.52 | 7.85 |
| | MAE | 0.62 | 0.46 | 0.26 |
| Vineyards | RMSE | 0.72 | 0.62 | 0.49 |
| | Bias | −0.15 | −0.12 | 0.00 |
| | Sigma | 0.42 | 0.36 | 0.29 |
| | R$^2$ | 0.651 | 0.675 | 0.692 |
| | RRMSE | 26.01 | 19.66 | 12.26 |
| | MAE | 0.59 | 0.50 | 0.33 |
| Pomegranates | RMSE | 0.48 | 0.44 | 0.50 |
| | Bias | −0.03 | −0.02 | 0.07 |
| | Sigma | 0.25 | 0.23 | 0.26 |
| | R$^2$ | 0.764 | 0.802 | 0.837 |
| | RRMSE | 12.96 | 10.96 | 14.30 |
| | MAE | 0.39 | 0.35 | 0.44 |
| Overall | RMSE | 0.69 | 0.56 | 0.45 |
| | Bias | −0.18 | −0.13 | −0.02 |
| | Sigma | 0.43 | 0.35 | 0.28 |
| | R$^2$ | 0.641 | 0.710 | 0.707 |
| | RRMSE | 24.85 | 16.29 | 10.51 |
| | MAE | 0.56 | 0.45 | 0.32 |

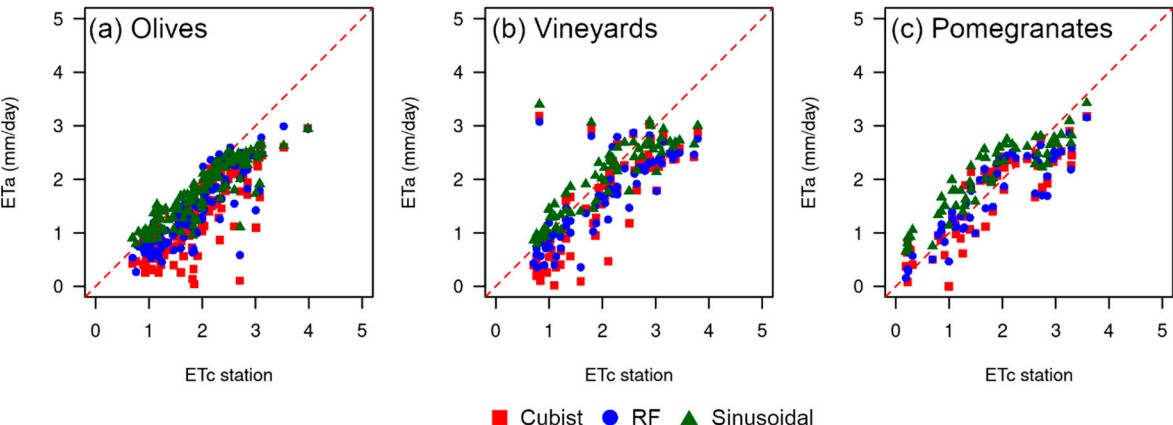

**Figure 10.** Predicted ETa at the 10 m resolution of the cubist, random forest, and sinusoidal model compared to the ETc station in (**a**) olives, (**b**) vineyards, and (**c**) pomegranates.

The SSEBop approach in arid ecosystems has been applied in quantification of irrigation in California [79], but it also might be a useful tool in ET estimations of semi-arid agroecosystems, such as the Copiapó valley. According to Anderson et al. [4], ET methods from vegetation indices tend to overestimate ET under stress conditions, showing higher crop demands before biomass can adjust. However, an estimation based on vegetation indices, such as the Sin method, might be useful as a primary approach in demand estimations in areas where water crop demand cannot be estimated using an ETc station with kc by calendar, and used in other areas of the valley as well. Furthermore, a spatial ET estimation based on Sentinel-2 frequency and spatial resolution would improve water demand quantifications in semiarid ecosystems such as the Copiapó valley, where groundwater demand is under pressure [29]. Monitoring these ecosystems will be crucial in order to minimize and prevent future conflicts over water in arid and semiarid climates, where higher water requirements will increase in the future [80,81]. It should be considered that the topographic effect can bring noise to NDVI retrieval, especially for these areas. According to previous studies, the topographic effect can be reduced by band ratios, due to the spectrum similarity between the NIR and visible bands [82,83]. About the limitations of the approach, our study used non-supervised areas for calibration of the LST models, using images of the whole study area instead of a selection of areas with vegetation, bare soil, or other surfaces. Besides, the models were evaluated during the seasonal variation of the agricultural vegetation; thus, they can perform differently in non-agricultural vegetation or in non-irrigated agriculture.

## 5. Conclusions

In this study, we evaluated three models to estimate and downscale LST using Sentinel-2 images and remote sensing indices as predictors, comparing them with Landsat-8 LST as the training data. The results of the LST predictions showed that the best model to downscale LST was a sinusoidal model, which showed the lowest RMSE of 3.97 K at 100 m and 3.4 K in 10 m, and with the highest correlation coefficients. The machine learning analysis showed that the variable with the greatest importance in predicting LST was Sentinel-2 band 9, as it was included in the majority of internal model conditions and prediction rules. These models were applied in ETa estimation using the operational surface energy balance method (SSEBop) with the downscaled LST, showing that the RF and Sin models are useful in estimating ETa in a semi-arid region. On the contrary, the Cub model was not reliable across space, not in the ETa predictions overall nor for the olives, vineyards, and pomegranates compared to Sin and RF.

This approach shows an advantage of the Sin model, which relies on NDVI and an equation related to the day of the year, and not on a set of other predictors. Therefore, a Sin model approach makes it possible to predict LST using previous date matches of Landsat

and Sentinel, without training a dataset that might vary between locations. The RF model showed the best overall performance estimating ET compared to all ETc stations, but a Sin model showed a similar performance to RF with the lowest RMSE in ETa in comparison to ETc overall and in olives, vineyards, and pomegranates. Future research should focus on improvements in Kc in situ measurements using ETa stations instead of Kc values from calendar, also testing the model quantification in irrigation scheduling considering the soil water content, saline stress, and plant ecophysiological variables.

Finally, this study contributes to estimating water demand in a semi-arid region by providing ETa maps at higher temporal and spatial resolutions and are reliable for crop water requirements and irrigation scheduling compared to ETc calculated with the kc values from the calendar.

**Supplementary Materials:** The following are available online at https://www.mdpi.com/article/10.3390/rs13204105/s1. Table S1. Statistics of Atmospheric inputs over Landsat 8 series. Figure S1. Mean, Standard Deviation and Coefficient of Variation of Aerosol Optical Thickness (AOT) retrieved from Sen2Cor for all images during study period over Copiapó valley. Figure S2. Mean, Standard Deviation and Coefficient of Variation of Water Vapor (WV) retrieved from Sen2Cor for all images during study period over Copiapó valley. Figure S3. Coefficient of determination ($R^2$) between Illumination and Sentinel 2 Bands, NDVI during study period in Copiapó valley.

**Author Contributions:** Conceptualization, L.A.R.R., I.M.-L., C.M., R.F. and C.E.-A.; Data curation, L.A.R.R., I.M.-L., F.C., C.M. and R.F.; Formal analysis, L.A.R.R., I.M.-L., F.C. and C.E.-A.; Funding acquisition, C.M., R.F. and C.E.-A.; Investigation, L.A.R.R., I.M.-L., F.C., C.M., R.F. and C.E.-A.; Methodology, L.A.R.R., I.M.-L., F.C., C.M. and R.F.; Project administration, C.M., R.F. and C.E.-A.; Resources, L.A.R.R., I.M.-L., F.C., C.M., R.F. and C.E.-A.; Software, L.A.R.R., I.M.-L. and F.C.; Supervision, I.M.-L., C.M., R.F. and C.E.-A.; Validation, L.A.R.R., I.M.-L., F.C., C.M., R.F. and C.E.-A.; Visualization, L.A.R.R. and F.C.; Writing–original draft, L.A.R.R.; Writing–review and editing, L.A.R.R., I.M.-L., F.C., C.M., R.F. and C.E.-A. All authors have read and agreed to the published version of the manuscript.

**Funding:** This work was funded by Chile's National Agency of Research and Development (ANID) [FONDEF project number IT18I0022].

**Acknowledgments:** We thank JWZ and SJV for proof reading the manuscript.

**Conflicts of Interest:** The authors have declared that no competing interest exist.

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
