# Peer review of "Determining Actual Evapotranspiration Based on Machine Learning and Sinusoidal Approaches Applied to Thermal High-Resolution Remote Sensing Imagery in a Semi-Arid Ecosystem"

_remotesensing, doi:10.3390/rs13204105_

Round 1
Reviewer 1 Report
Dear authors,
Please find the attached file to address the comments and suggestions in order to improve the manuscript quality for further consideration.

Author Response
Dear reviewer, we would like to submit our changes to your comments. We included new sections in methodology about the data process before the calibration, supporting the retrieval of surface reflectance and land surface temperature data. We included supplementary figures as well and an update in references of equations as was suggested by a reviewer.
Please see the attachment
Thank you for your time and consideration of this manuscript,

Reviewer 2 Report
Land surface temperature retrieved from Landsat-8 was downscaled from 100m to 20m by three methods, which was then used to estimate actual evapotranspiration based on SSEBop model. In general, this paper’s topic is very suitable for Remote Sensing. I have two main concerns. (1) The accuracy of Sin model for LST downscaling is superior to the accuracy of other two machine learning models. Why? A large number of studies have shown the superiority of machine learning models in accuracy. Why has a simple model using only NDVI as input achieved the best accuracy? Are you sure you have applied these two machine learning models well? (2) The performance of ET estimation was evaluated based on ETc = ETo ∙kc on monthly scale. I don’t think it is accurate enough to be treated as reference. Also, I am not sure whether the parameter kc here is the same as the kc in FAO56 file. Monor comments: (1) The sample number of Eq. (4) and (5) is 18. Please clarify the sample number of Eq. (3). I suppose a、b in Eq. (2) are the same as c、d in Eq. (3). (2) Why is the resolution of Landsat-8 100m rather than 30m? (3) LST was retrieved from Landsat-8 using Single-channel algorithm. What about the accuracy of this algorithm? Why do you choose this algorithm because there are several other algorithms? (4) Please clarify ETf in SSEBop model. What is its physical meaning? (5) In Figure 4 take vineyards as an example. It seems that all models have overestimated LST. This is not reasonable. (6) Although the LST was downscaled to 10m resolution, your ET estimation in Figure 7 shows little details. I don’t see any improvement in the resolution of ET estimation. (7) The language needs much improvement.Author Response
Dear reviewer, we would like to submit our changes to your comments. We included new sections in methodology about the data process before the calibration, supporting the retrieval of surface reflectance and land surface temperature data. We included supplementary figures as well and an update in references of equations as was suggested by a reviewer.
Please see the attachment
Thank you for your time and consideration of this manuscript

Reviewer 3 Report
The manuscript is focused on the determining of actual evapotranspiration via remote sensing methods in the study area. The methods are clearly described and the main benefit is the testing methods for Land surface temperature (LST) estimation via three different methods. Actual evapotranspiration is calculated based on several "constant" values (e.g. aerodynamic resistance, crop coefficients) therefore the spatial distribution of ETa is more or less similar to spatial distribution of LST. It is unclear for me how the ET0 values via Penman-Monteith eq. have been calculated in spatial resolution.
I have some questions/recommendations for clarifying:
- Has the irrigation system in the study area been active during the time of analyzed period ? Please to add this information to manuscript.
- How have the ET0 data (P-M eq.) been calculated in spatial resolution, please to describe in more details.
- In the discussion should be more information about the limitation of this method due to many simplifying assumptions (some processes have been converted to constant values)
Author Response
Dear reviewer, we would like to submit our changes to your comments. We included new sections in methodology about the data process before the calibration, supporting the retrieval of surface reflectance and land surface temperature data. We included supplementary figures as well and an update in references of equations as was suggested by a reviewer.
Please see the attachment
Thank you for your time and consideration of this manuscript

Round 2
Reviewer 1 Report
I want to thank the authors for addressing previous comments thoroughly and improving the manuscript in a well structured manner.
Reviewer 2 Report
All my concerns have been answered well. The submission is now suitable for publication.